

# Hybrid adaptive control for variable-speed variable-pitch wind energy systems using general regression neural network

Xiuxing Yin[1]

[1] State Key Laboratory of Fluid Power Transmission and Control, Zhejiang University, Hangzhou 310027, China.

*Correspondence to*: Xiuxing Yin (calminder@126.com)

**Abstract.** This paper presents a novel hybrid adaptive control approach for the variable speed-variable pitch (VS-VP) semi-direct driven WECS by combining pitch control with variable generator torque regulation in different operating regions. A general regression neural network (GRNN) is employed to derive the reference commands of generator torque and pitch angle from the real-time signals of generator power and speed. Furthermore, a fast and effective nonlinear PID pitch controller is

presented to track the reference command of pitch angle in the full load region. The proposed GRNN based hybrid adaptive control strategies have been developed and validated using comparative simulations. This study shows that the proposed method is much faster, more accurate and effective than conventional linear control approach.

## 1. Introduction

Nowadays, wind energy conversion systems (WECSs) have received considerable attention and enjoyed increasing interest

around the world due to their technological enhancement, significant cost reduction and improved conversion efficiency. As WECSs are augmented in size and power rating, control specifications and strategies become more and more demanding. Increasingly, control systems have been expected not merely to keep the WECS within its safe operating regions but also to improve efficiency and quality of power conversion.

WECSs can be programmed and controlled to work in different modes of operation. Fixed-speed (FS), variable-speed (VS),

fixed-pitch (FP) and variable-pitch (VP) are the commonest ones. Since WECSs work under different conditions, these modes of operation are usually combined to attain the control objectives over the full range of operational wind profiles. In the operation mode of FS-FP, the generator speed is locked according to the power frequency and the torque characteristic cannot be modified. As a result, WECS has low conversion efficiency, poor power quality, high-frequency loads and vibration. The FS-VP mode of operation means that maximum power conversion is attainable only at a single wind speed. Therefore,

conversion efficiency below rated wind speed cannot be optimized. The benefits ascribed to VS-FP operation mode are larger energy conversion efficiency, dynamic loads alleviation and power quality enhancement. Maximum efficiency conversion can be achieved below rated wind speed. In the scheme of VS-VP operation mode, WECS is controlled to operate at variable speed and fixed pitch below the rated wind speed and at variable pitch above rated wind speed. Variable-speed operation enables maximum energy exploiting under low wind speed while variable-pitch operation enables efficient power regulation above the



rated wind speed. Therefore this mode of operation not only achieves the ideal power curve, but also alleviates transient loads under high wind conditions.

A large amount of previous researches have focused on the maximum wind power extraction by regulating generator torque for variable-speed (VS) WECS under rated power. Adaptive neuron-fuzzy inference system (Meharrar, A., et al., 2011) and generic maximum power point tracking controller (Narayana, M., et al. and Messai, A., et al., 2012) are proposed to exploit the maximum power from wind; Takagi–Sugeno–Kang (TSK) fuzzy model combined with genetic algorithms (GA) and recursive least-squares (LS) optimization methods are used in (Calderaro, V., et al., 2008, Galdi, V., A. Piccolo and P. Siano, 2009) for maximum energy extraction from variable speed wind turbines; A Wilcoxon radial basis function network (WRBFN) with hill-climb searching (HCS) MPPT strategy using a back-propagation learning algorithm with modified particle swarm optimization (MPSO) regulating controller is designed in (Lin, W. and C. Hong, 2010) for a PMSG to achieve the maximum power point tracking. Other researchers paid more attention on the variable-pitch control for WECS above the rated wind speed. A radial basis function (RBF) neural network based PI controller optimized by particle swarm optimization (PSO) evolutionary algorithm is proposed for collective pitch control (CPC) of a 5MW wind turbine in (Poultangari, I. et al. 2012); An artificial neural network-based pitch angle controller for wind turbines is used in (Yilmaz, A.S. and Z. Özer et al., 2009) to regulate output power and prevent overloading.

In (Taher, S.A. et al., 2013), an optimal gain scheduling controller (GSC) for a variable-pitch variable-speed wind energy conversion system (VS-WECS) is presented, aiming to control the output electrical power as well as the shaft speed in the above-rated wind speeds. However, the optimal gain scheduling controller cannot be used to exploit the maximum wind power from wind below-rated wind speeds, and it is difficult to measure or obtain the wind data, so the dynamic accuracy of the controller should be further considered.

In this paper, we present a hybrid adaptive control approach for the VS-VP half-direct driven WECS by combining pitch angle control with variable generator torque regulation in different operating regions. Under the rated wind speed, WESCs are operated at variable speed and fixed pitch with the aim of maximizing wind power extraction and minimizing the transient loads. When wind speed is higher than its rated value, both the generator torque characteristic and the pitch angle are controlled simultaneously in the aim of high frequency load mitigation and power smoothing. Reference commands of generator torque and pitch angle are derived from the general regression neural network (GRNN) independently and simultaneously. In the GRNN model, the input variables are real-time signals of generator power and speed, the output reference commands of generator torque and pitch angle are then soft-calculated by the interconnected artificial neurons and particular learning algorithm. Further, a fast and effective nonlinear PID pitch controller is proposed to track the pitch angle reference command in the full load operation region. The multi-variable hybrid adaptive control approach has the low memory occupancy, fast soft computing speed, and is independent from the wind profiles or wind speed estimation (Tian, L., Q. Lu and W. Wang, 2011). Hence, the control reliability and accuracy can be guaranteed, capabilities of VS-VP WECS will be fully exploited.



## 2. Modelling of WECS

The wind energy conversion system (WECS) studied in this paper is shown in Fig.1, where the horizontal axis and variable-speed variable-pitch wind turbine is coupled to the high-speed shaft PMSG through a gear box, and the PMSG is connected with the power converters. Major components of the WECS are the wind turbine, the drive-train, and the PMSG. Main parameters of this WECS are shown in the appendix.

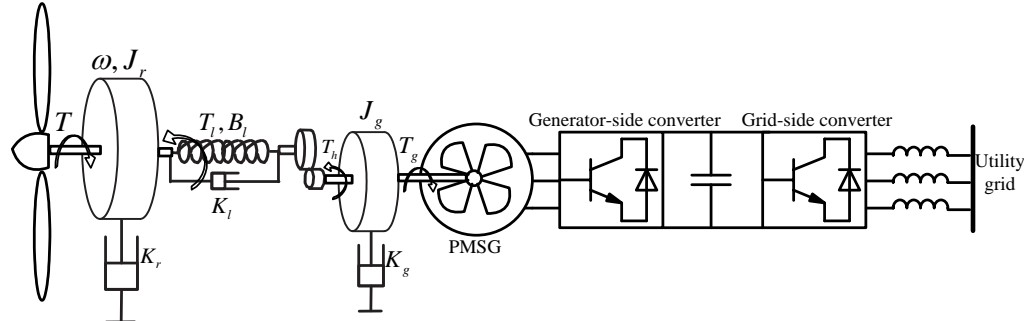

Fig.1 Schematic diagram of the half-direct driven WECS

### 2.1 Characteristics of wind turbine

For a wind turbine, the output mechanical power and aerodynamic torque extracted from the wind turbine can be described as

$$P = \frac{1}{2}\pi\rho R^2 v^3 C_p(\lambda, \beta) \tag{1}$$

$$T = \frac{1}{2}\pi\rho R^3 v^2 C_p(\lambda, \beta) / \lambda \tag{2}$$

where $P$ is the mechanical power (MW), $T$ is the aerodynamic torque of wind turbine (MNm), $v$ denotes the wind speed at the center of the rotor ($m/s$), $R$ is the radius of wind rotor ($m$), $\rho$ is the air density ($kg/m^3$), $\lambda$ is the tip speed ratio (TSR), $\beta$ is the pitch angle, $C_P$ is the power coefficient, which is the nonlinear function of the tip speed ratio and the pitch angle.

The tip speed ratio $\lambda$ (TSR) can be described as

$$\lambda = \frac{\omega R}{v} \tag{3}$$

Eq.(3) denotes that the TSR is the ratio between the tip speed and the upstream wind speed of the wind turbine, where $\omega$ is the wind rotor angular velocity (rad/sec).

There exists an optimum value of TSR $\lambda_{opt}$ that leads to the maximum power coefficient $C_{pmax}$ for a wind turbine. From (1) and (3), we get





$$P_{\max} = \frac{\pi \rho R^5 C_{p\max}(\lambda, \beta)}{2\lambda_{opt}^3} \omega_{opt}^3 \qquad (4)$$

When the power coefficient $C_p$ reaches its maximum value $C_{pmax}$, the optimum aerodynamic torque $T_{opt}$ can be described as

$$T_{opt} = \frac{\pi \rho R^5 C_{p\max}(\lambda, \beta)}{2\lambda_{opt}^3} \omega_{opt}^2 \qquad (5)$$

Pitch control subsystem is always an alternative to power limitation for MW scale wind turbine above rated wind speed.
The pitch actuator is a nonlinear servo hydraulic or electromechanical mechanism that generally rotates the blades around their longitudinal axes.

The pitch system can be modeled as a first-order dynamic system with saturation in the amplitude and derivative of the control signal. The dynamic characteristic of the pitch actuator operating in its linear region can be described by the following differential equation

$$\dot{\beta} = -\frac{1}{\tau}\beta + \frac{1}{\tau}\beta_{ref} \qquad (6)$$

Where $\beta$ and $\beta_{ref}$ are the actual and reference pitch angles, respectively. Typically, $\beta$ ranges from $-2°$ to $30°$, and varies at a maximum rate of $\pm10°$ /s. $\tau$ denotes the time constant.

**2.2 Modelling of the drive-train**

The power transmission from wind rotor to generator is done by the drive-train. In this paper, a two mass model as shown
in Fig. 1 is employed to model the drive-train. The dynamic response of the wind rotor driven at a speed $\omega$ by the aerodynamic torque $T$ can be described as

$$J_r \dot{\omega} = T - T_l - K_r \omega \qquad (7)$$

where $J_r$ denotes the wind rotor inertia, $K_r$ is the wind rotor external damping. The low-speed shaft torque $T_l$ acts as the braking torque on the wind rotor, and it results from the torsion and friction effects due to the difference between $\omega$ or and $\omega_l$
(Boukhezzar, B. and H. Siguerdidjane, 2011):

$$T_l = B_l(\theta - \theta_l) + K_l(\omega - \omega_l) \qquad (8)$$

where $\theta$ is the rotor-side angular deviation, $\theta_l$ denotes the low speed shaft angular deviation, $\omega_l$ is the angular speed of low speed shaft. $B_l$ and $K_l$ are the low speed shaft stiffness and damping, respectively.

The generator is driven by the high speed shaft torque $T_h$ and braked by the generator electromagnetic torque $T_g$:
$$J_g \dot{\omega}_g = T_h - K_g \omega_g - T_g \qquad (9)$$

where $K_g$ is the external damping of generator, $J_g$ denotes the generator inertia.





Assuming an ideal gearbox with transmission ratio $N$, the torque and speed relation between high speed shaft and low speed shaft is calculated as

$$N = \frac{T_l}{T_h} = \frac{\omega_g}{\omega} \tag{10}$$

where $\omega_g$ is the angular velocity of the high shaft or generator.

Using Eqs (9) and (10) and transferring the generator dynamics to the low speed shaft, the generator dynamics can be written as

$$N J_g \dot{\omega}_g = T_l - N K_g \omega_g - N T_g \tag{11}$$

When a perfect rigid low speed shaft is assumed, the single-mass model of the drive train can be established using Eqs (7) (10) and (11) as

$$J \dot{\omega} = T - K \omega - T_e \tag{12}$$

where

$$J = J_r + N^2 J_g$$
$$K = K_r + N^2 K_g$$
$$T_e = N T_g$$

## 2.3 Modelling of PMSG

Permanent magnet generators (PMSGs) are appropriate and widely employed for WECS because they are self-excited,
brushless, and require little maintenance.

PMSGs are usually modeled assuming uniform distribution of stator 3-phase windings, electrical and magnetic symmetry, unsaturated magnetic circuit and negligible iron losses. Model of the generator can be obtained by the Park transformation from the reference coordinate frame ($a$, $b$, $c$) to ($d$, $q$) (Tan, K. and S. Islam, 2004), where the PMSG can be modeled as

$$
\begin{aligned}
u_d &= R i_d + L_d \frac{d i_d}{dt} - \phi_q \omega_s \\
u_q &= R i_q + L_q \frac{d i_q}{dt} + \phi_d \omega_s
\end{aligned}
\tag{13}
$$

where $u_{d,q}$ are the stator voltages in the ($d$, $q$) reference coordinate frame, $L_d, L_q$ are the $d$, $q$ axis stator inductances. $R$ is the stator winding resistance, $i_{d,q}$ are the currents in the $d$, $q$ axes, $\Phi_{d,q}$ are the flux linkages, $\omega_s$ denotes the angular velocity of the stator voltage.

The flux linkages $\Phi_{d,q}$ can be described in terms of the stator currents and the magnetic flux as following:



$$\phi_d = L_d i_d + \phi_m$$
$$\phi_q = L_q i_q \tag{14}$$

where, $\Phi_m$ is the constant flux linkage of the permanent magnets.

After substituting Eq.(14) into Eq.(13), the PMSG model becomes

$$u_d = R i_d + L_d \frac{di_d}{dt} - L_q i_q \omega_s$$
$$u_q = R i_q + L_q \frac{di_q}{dt} + \left( L_d i_d + \phi_m \right) \omega_s \tag{15}$$

The PMSG electromagnetic torque $T_g$ of $p$-pole machine is obtained as

$$T_g = p \left( \phi_d i_q - \phi_q i_d \right) = p \left[ \phi_m i_q + \left( L_d - L_q \right) i_d i_q \right] \tag{16}$$

where, $p$ is the number of pole pairs.

For a non-salient-pole PMSG, if the permanent magnets are uniformly mounted on the rotor surface, then $L_d$ and $L_q$ may be approximately equal and $T_g$ can be obtained from the following equation

$$T_g = p \phi_m i_q = K_g i_q \tag{17}$$

where $K_g$ is the generator torque constant.

From Eq.(17), the electromagnetic torque $T_g$ is proportional to the current $I_q$, hence, the electromagnetic torque can be varied by controlling the PMG current. Then, the electrical power can be described as

$$P_g = \omega_g T_g = P \eta \tag{18}$$

where $\eta$ denotes the power transmission efficiency from the wind turbine to generator and can be viewed as a constant between 0 and 1.

## 3. Operating regions and control strategies of WECS

WECSs are always controlled and operated in different modes of operation where control objectives and strategies are different from each other and should be illustrated in detail, respectively.

**3.1 Operating regions and control objectives for VS-VP WECS**

There are mainly four regions of operation for VS-VP WECS and these operating regions can be delimited by the cut-in ($v_{min}$) and cut-out ($v_{max}$) wind speeds. WECS remains stopped beyond the two limits. Below cut-in wind speed (region I), the available wind energy is too low to compensate for the operation costs and losses. Above the cut-out wind speed (region IV), WECS is shut down to prevent itself from structural overload.



In the partial load region (region II), the wind speed is upper than $v_{min}$ but lower than rated value $v_N$, and the available power is lower than rated power, therefore, the control objective in the region is to extract the maximum available wind power defined by Eq.(4) and minimizing the transient loads.

In the full load region (region III), the wind speed is upper than $v_N$, but lower than the cut-out ($v_{max}$). The main control
objective in this region is to limit the generated power $P_g$ around its rated value $P_{gN}$ to avoid overloading. In this region, the available wind power always exceeds the rated value. Therefore, WECS must be operated with power efficiency lower than $C_{pmax}$.

### 3.2 Control strategies in different operating regions

Control strategies are closely related with the aforementioned control objectives and will vary in different operating regions.
In the partial load region (region II), the power coefficient $C_{pmax}$ will be maximized in order to extract the maximum power from wind energy. The rotor angular velocity is then varied in proportion to the wind speed to maintain $\lambda=\lambda_{opt}$, whereas the pitch angle is kept constant at $\beta_o$.

For higher wind speed region (region III), control objective can be equivalent to keeping the rotor angular velocity and generator power at their rated values. Hence, this control strategy can be achieved by regulating the pitch angle and generator
torque simultaneously to maintain the captured generator power at its rated value $P_{gN}$.

Reference command of generator torque $T_{g,ref}$ can be deduced from the Eqs. (11), (12), (18) and can be mathematically described by the following equation

$$T_{gref} = \begin{cases} \left. \dfrac{P_g}{\omega_g \eta} - \dfrac{\left(K_r + N^2 K_g\right)\omega_g - \left(J_r + N^2 J_g\right)\dot{\omega}_g}{N^2} \right|_{\omega_g = N\sqrt[3]{\frac{2P_{max}\lambda_{opt}^3}{\pi\rho R^5 C_{pmax}}}} & v_{min} < v < v_N \\[4mm] \left. \dfrac{P_{gN}}{\omega_g \eta} - \dfrac{\left(K_r + N^2 K_g\right)\omega_g - \left(J_r + N^2 J_g\right)\dot{\omega}_g}{N^2} \right|_{\omega_g = \omega_{gN}} & v_N < v < v_{max} \end{cases} \tag{19}$$

where $\omega_{gN}$ denotes the rated angular velocity of generator.
Reference command of pitch angle will also be varied in different operating regions and can be described as the following equation using the Eqs.(1), (3), (10), (18).

$$\beta_{ref} = \begin{cases} \beta_o & v_{min} < v < v_N \\ f(P_g, \omega_g) & v_N < v < v_{max} \end{cases} \tag{20}$$

where $\beta_{ref}$ is a constant value $\beta_o$ in the partial load region and is the nonlinear function of the generator angular velocity $\omega_g$ and generator power $P_g$ in the full load region.





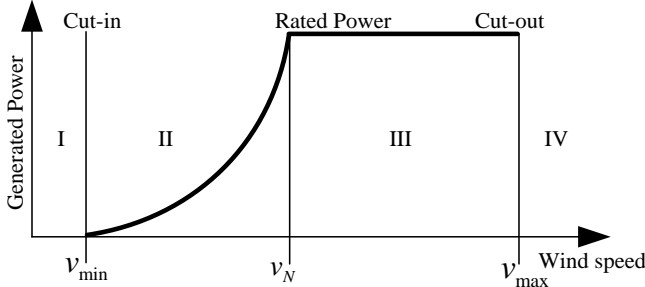

Fig.2. Operating regions for VS-VP WECS

## 4. GRNN based hybrid adaptive control

Neural network control approaches do not require precise mathematical model, structure parameters or dynamic characteristics of controlled system. They can deduce the control commands from input variables by their abilities of learning and generalization.

In this paper, a generalized regression neural network (GRNN) is employed to extract the reference commands of generator torque $T_{gref}$ and pitch angle from the input real-time measurements of generator angular velocity $\omega_g$ and power $P_g$.

Generalized regression neural network (GRNN) has strong ability of nonlinear mapping, flexible network architecture, good characteristics of fault tolerance and robustness. GRNN has the ability of approximating any nonlinear function with high accuracy and good generalization capability even in the condition of few input samples. Based on the established statistical principles and converges with an increasing number of samples asymptotically to the optimal kernel regression surface, GRNN does not require an iterative training procedure and is particularly advantageous with sparse data in a real-time environment where the regression surface is instantly defined everywhere.

As described in Eq.(21), there exists a conditional mean between the dependent stochastic variable $y$ and independent stochastic variable $x$ ,when assuming that the observed value of $x$ is $X$ and the joint probability density function is $f(x,y)$.

$$\hat{Y} = E(y/X) = \frac{\int_{-\infty}^{\infty} yf(X,y)dy}{\int_{-\infty}^{\infty} f(X,y)dy} \tag{21}$$

Where $\hat{Y}$ denotes the predicted output of $Y$ when $X$ is the input stochastic variable.

The joint density function $f(X,y)$ can then be evaluated by using the Parzen nonparametric estimation from a sample data set.



$$\hat{f}(X,y) = \frac{\sum_{i=1}^{n} \exp\left[-\frac{(X-X_i)^T(X-X_i)}{2\sigma^2}\right]\exp\left[-\frac{(X-Y_i)^2}{2\sigma^2}\right]}{n(2\pi)^{\frac{p+1}{2}}\sigma^{p+1}} \tag{22}$$

where $f(X,y)$ denotes the estimated value of $\hat{f}(X,y)$, $X_i$ and $Y_i$ are the observed sample dataset values of $x$ and $y$ respectively; $n$ is the sample dataset size; $p$ is the dimension of $x$; $\sigma$ is the smoothing factor.

When replacing $f(X,y)$ with $\hat{f}(X,y)$ in the Eq.(21), we get the predicted output value of $Y$.

$$\hat{Y}(X) = \frac{\sum_{i=1}^{n} \exp\left[-\frac{(X-X_i)^T(X-X_i)}{2\sigma^2}\right]\int_{-\infty}^{\infty} y\exp\left[-\frac{(X-Y_i)^2}{2\sigma^2}\right]dy}{\sum_{i=1}^{n} \exp\left[-\frac{(X-X_i)^T(X-X_i)}{2\sigma^2}\right]\int_{-\infty}^{\infty} \exp\left[-\frac{(X-Y_i)^2}{2\sigma^2}\right]dy} \tag{23}$$

where $\hat{Y}(X)$ is the weight mean of the observed sample data set value $Y_i$, and the weight factor of $Y_i$ is the square index of the Euclid distance between $X_i$ and $X$.

The smoothing factor $\sigma$ influences the GRNN performance and should be given an appropriate value so that the GRNN has good generalization ability, good approximation performance.

According to the aforementioned kernel regression theory, GRNN can be organized using a input layer, a pattern layer, a summation layer, a output layer. When the input variables are $[x_1, x_2, \cdots, x_n]^T$, the output variables are $[y_1, y_2, \cdots y_k]^T$.

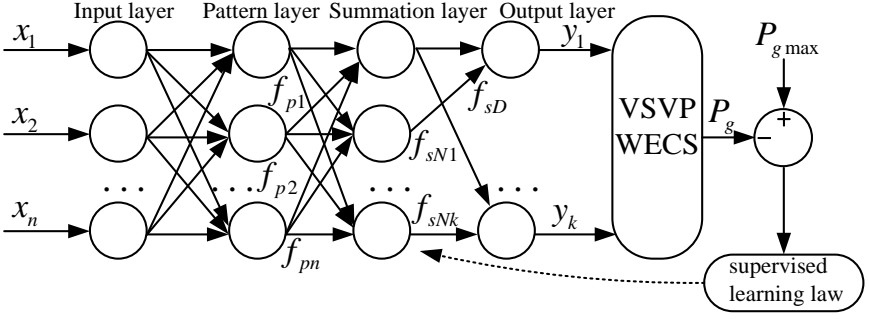

Fig. 3 The schematic representation of the GRNN

As shown in Fig.3, the input layer receives the input signals of GRNN. The number of nodes in the input layer is equal to

the input vector dimension of sample dataset. The pattern layer possessed a nonlinear transformation applied on the dataset from input space to the pattern space. The most popular transfer function of the pattern layer can be descried as

$$f_{pi} = \exp\left[-\frac{(X-X_i)^T(X-X_i)}{2\sigma^2}\right], i=1,2,...,n \tag{24}$$

where $X$ is the input variable of the network, $X_i$ is the corresponding sample dataset value of the $i$ node.

The summation layer executes the sum operation. The transfer function of the summation layer can be expressed as the

summation of the transfer function of pattern layer.




$$f_{sNj} = \sum_{i=1}^{n} y_{ij} f_{pi}, \, j = 1, 2, ..., k \tag{25}$$

where $y_{ij}$ denotes the connection weight between the $i$ node in the pattern and the $j$ node in the summation layer.

When $y_{ij} = 1$, the Eq.(25) can be simplified as

$$f_{sD} = \sum_{i=1}^{n} f_{pi} \tag{26}$$

The number of nodes in the output layer is equal to the output vector dimension of sample dataset. The predicted output of the node $j$ can be defined as

$$y_j = \frac{f_{sNj}}{f_{sD}}, \, j = 1, 2, ..., k \tag{27}$$

In the output layer, the nodes are represented by a GRNN individual output.

As described in Eq.(28), the root mean square error (RMSE) can be used as the termination criterion and performance indices

of the GRNN. The optimal value of smoothing factor can be obtained in a supervised learning and training process

$$RMSE = \frac{1}{n} \sum_{i=1}^{n} \left[ \hat{Y}_i(X_i) - Y_i \right]^2, \, i = 1, 2, ..., n \tag{28}$$

where $\hat{Y}(X_i)$ denotes the predicted output when $X_i$ is the input variable of the GRNN.

In this paper, the input variables of the GRNN are real-time measurements of generator angular velocity $\omega_g$ and generator power $P_g$, and the output variables are the reference commands of generator torque $T_{gref}$ and pitch angle $\beta_{ref}$. Moreover, the

value of dimension $n$ and $k$ are 2 respectively.

The cost function deduced from the RMSE criterion can be described as

$$E_p = \frac{1}{n} \sum_{i=1}^{n} \left[ P_{gref} - P_g \right]^2 \tag{29}$$

The weight factors can then be updated by supervised learning law which can be expressed as

$$W(m+1) = W(m) - \xi(m) \frac{\partial E_p}{\partial W(m)} \tag{30}$$

where $W(m)$ is the weight factor vector in the pattern or summation layers. $\xi$ denotes the learning rate. $m$ is the learning step.

GRNN model approximates the nonlinear functions described in Eqs.(19) (20), and computes the reference commands of generator torque $T_{gref}$ and pitch angle $\beta_{ref}$ automatically from the input variables of generated power $P_g$ and the angular velocity of generator $\omega_g$ in different operating regions of WECS.




As shown in Fig.4, the Hybrid adaptive control system consists of the GRNN model, nonlinear PID pitch controller, and a coordinate translator used for field-orientation control. Real-time measurements of generated power $P_g$ and the angular velocity of generator $\omega_g$ are put into the GRNN model as input variables and the GRNN generates the reference commands of generator torque $T_{gref}$ and pitch angle $\beta_{ref}$ as the input control variables to the torque or pitch control systems.

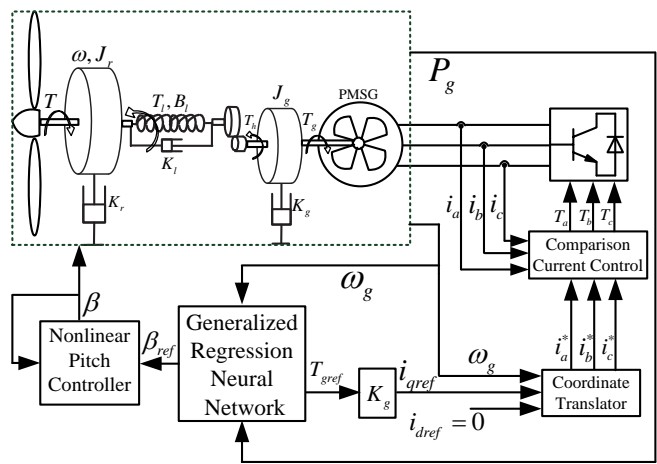

Fig. 4 Block diagram of GRNN based hybrid control

## 5. Novel nonlinear PID pitch controller

A novel nonlinear PID pitch controller is proposed in this paper to precisely control the pitch system. The nonlinear PID pitch controller has strong robustness of avoiding external perturbations and parameter variations as compared to conventional

PID controller. Furthermore, this controller also owns the advantages of compactness, easy implementation.

This novel controller comprises a nonlinear stator error feedback controller (NLSEF) and two nonlinear tracking differentiators (TD), *i.e.* TDI and TDII. The first nonlinear tracking differentiator (TDI) can filter the rough pitch angle reference command generated from GRNN model into a smooth and suitable differential signal for the nonlinear PID pitch controller. The second one can reconstruct the output pitch angle signal and obtain its differential feedback signal.

As shown in Fig.5, The pitch angle reference command is sent to the first nonlinear tracking differentiator (TDI) for nonlinear mapping and two signals $z_{11}$ and $z_{12}$ are then extracted from the TDI, where $z_{11}$ follows the track of the pitch angle reference command and $z_{12} = \dot{z}_{11}$. The output pitch angle signal is sent to the second nonlinear tracking differentiator (TDII) and two signals $z_{21}$ and $z_{22}$ are then extracted, where $z_{21}$ follows the track of the output pitch angle signal and $z_{22} = \dot{z}_{21}$. Therefore, the signal error $\varepsilon_0$, integral error $\varepsilon_1$, and differential error $\varepsilon_2$ are generated for the calculation of pitch

angle which can be deduced by the nonlinear combination and function of the signal errors $\varepsilon_0$, $\varepsilon_1$, $\varepsilon_2$.

The signal errors $\varepsilon_0$, $\varepsilon_1$, $\varepsilon_2$ in this nonlinear pitch controller can be mathematically described as



$$\begin{cases} \varepsilon_0 = z_{11} - z_{21} \\ \varepsilon_1 = \int_0^t (z_{11} - z_{21})\, dt \\ \varepsilon_2 = z_{12} - z_{22} \end{cases} \tag{31}$$

The two nonlinear tracking differentiators (TDI and TDII) can be mathematically described as

$$\begin{cases} \dot{z}_{12} = -r_1 g \left[ z_{11} - \beta_{ref} + \dfrac{z_{12}|z_{12}|}{2r_1}, \theta_1 \right] \\ \dot{z}_{22} = -r_2 g \left[ z_{21} - \beta + \dfrac{z_{22}|z_{22}|}{2r_2}, \theta_2 \right] \end{cases} \tag{32}$$

Output pitch angle can then be obtained from the nonlinear stator error feedback controller as

$$\beta = k_p f(\varepsilon_0, \alpha_0, \delta_0) + k_i f(\varepsilon_1, \alpha_1, \delta_1) + k_d f(\varepsilon_2, \alpha_2, \delta_2) \tag{33}$$

Nonlinear functions can be expressed as

$$g(x, \theta) = \begin{cases} \operatorname{sgn}(x) & |x| \ge \theta \\ x/\theta & |x| < \theta \end{cases} \tag{34}$$

$$f(\varepsilon, \alpha, \delta) = \begin{cases} |\varepsilon|^{\alpha} \operatorname{sgn}(\varepsilon) & |\varepsilon| > \delta \\ \varepsilon/\delta^{1-\delta} & |\varepsilon| \le \delta \end{cases} \tag{35}$$

where $k_p\, k_i\, k_d$ are the tunable gain factors, $r_1$ and $r_2$ are the system coefficients of nonlinear tracking differentiators, respectively.
10  $\delta_0\, \delta_1\, \delta_2$ are the linear interval indexes of the nonlinear function $f(\varepsilon, \alpha, \delta)$. $\alpha_0\, \alpha_1\, \alpha_2$ are the nonlinear interval indexes of the nonlinear function $f(\varepsilon, \alpha, \delta)$, respectively. If $0 < \alpha < 1$, then the nonlinear function $f(\varepsilon, \alpha, \delta)$ has the characteristic of "high error, small gain factors; low error, large gain factors", while if $\alpha > 1$, then the nonlinear function $f(\varepsilon, \alpha, \delta)$ has the characteristic of "high error, large gain factors; low error, small gain factors". The characteristics indicate that the nonlinear function $f(\varepsilon, \alpha, \delta)$ has good ability of suppressing signal error and perturbation. $\theta_1\, \theta_2$ are the linear interval indexes of the nonlinear function
15  $g(x, \theta)$, respectively.



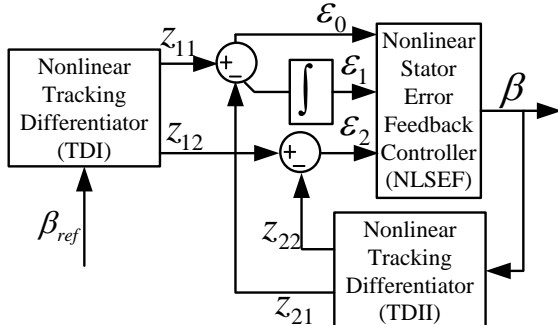

Fig. 5 Block diagram of the novel nonlinear PID pitch controller

## 6. Test results and discussions

The effectiveness and capability of the proposed control approach have been verified through in-house experiments in a 30 kW wind energy simulator system in MATLAB software. The wind turbine simulator is employed to simulate the wind turbine dynamics by exerting nearly realistic wind turbine torque to the gearbox and generator system. The gearbox was designed with a bevel geared transmission in order to save space for testing. The electrical load is used to dissipate the generated electrical power from the generator and hence to act the external loading to the generator. The control algorithm proposed in this paper was programmed and tested in MATLAB software. Comparisons and analysis of the performances between the proposed hybrid adaptive control method and conventional linear control for VS-VP WECS as well as the novel nonlinear PID pitch controller and conventional PID pitch controller for pitch control are also given in this paper.

In Fig. 6(a), a 5-min data set of turbulent wind was generated using the Class A Kaimal turbulence spectra. It has a mean value of 15 m/s at the hub height and a turbulence intensity of 25%.

According to Fig. 6(b), both the linear control and GRNN based torque controllers achieve an acceptable rotor angular velocity regulation in the partial load region. However, tracking performance of the conventional linear controller is poor, while the rotor angular velocity can be precisely controlled to track the optimum angular velocity by GRNN based torque controller.

As shown in Fig. 6(c), the generator torque from the linear controller is always under relatively big deviation to the generator torque reference command, while the generator torque derived from the GRNN tracks the reference command more accurately.

According to Fig. 6(d), the maximum generator power extraction can be achieved by the hybrid adaptive controller and the tracking error is relatively smaller while the generator power extracted from the linear controller has much more deviations from the maximum generator power.





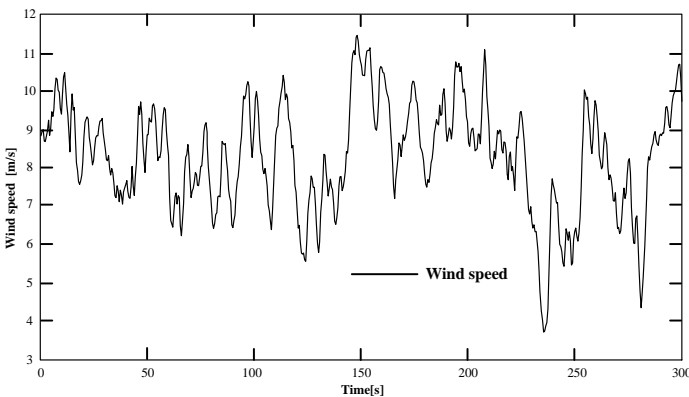

Fig. 6(a). Wind profile in the partial load region condition

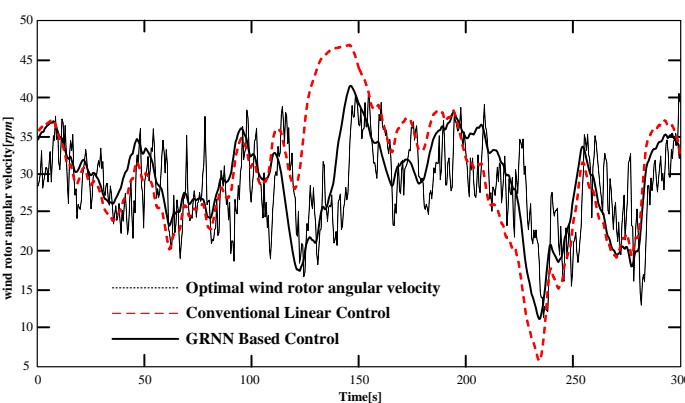

Fig. 6(b). Comparison of wind rotor angular velocity in the partial load region condition

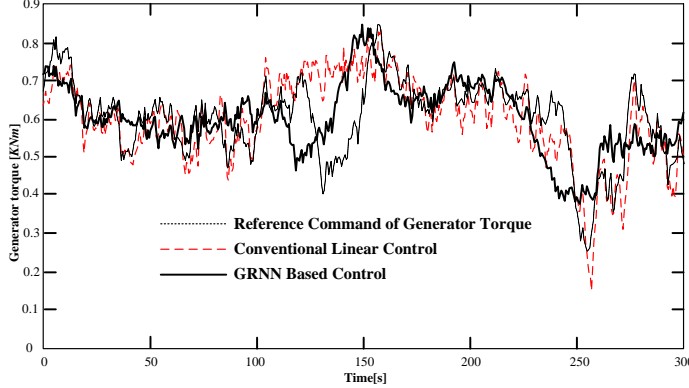

Fig. 6(c). Comparison of generator torque in the partial load region condition




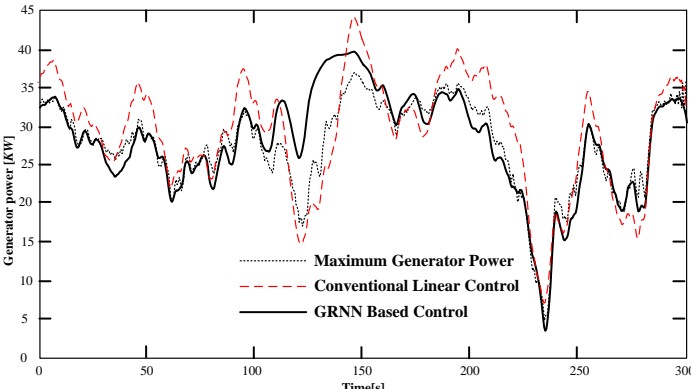

Fig. 6(d). Comparison of generator power in the partial load region condition

Wind profile shown in Fig. 7 (a) is employed to verify the performance of each controller in the full load region condition. This wind profile covers different range of wind speed and simulates the randomness of the actual wind velocity. The maximum,

minimum and average values of wind speed are 20m/s, 13m/s and 16m/s respectively.

In Fig. 7 (b), the pitch control performances of the hybrid adaptive controller and conventional linear controller are compared. The self-adaptive nonlinear PID pitch controller not only achieves a good tracking performance of the pitch angle reference command but also has much faster response than the conventional PID pitch controller.

Comparison of generator torque in the full load region condition is shown in Fig. 7 (c). The hybrid adaptive torque controller

has a better tracking performance of the reference command than the conventional linear controller and has a relatively less-turbulent range.

As shown in Fig. 7 (d), due to the fast response of the controller, the hybrid adaptive controller achieves good generator power regulation. The generator power is better maintained to its rated value 30kW using the hybrid adaptive controller as compared to the linear controller where there exists a relatively large power deviation from the rated value. Therefore, the

combination of the generator torque and pitch control signals in the full load region condition leads to a multivariable approach that can achieve the double objectives of regulating the generator torque and electrical power.

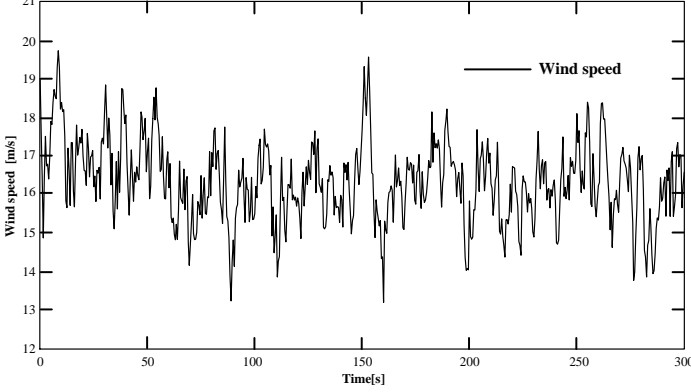

Fig. 7 (a). Wind profile in the full load region condition



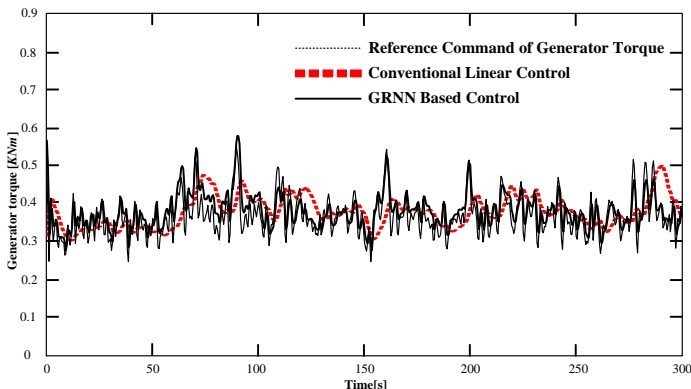

Fig. 7 (b). Comparison of pitch angle in the full load region condition

Fig. 7 (c). Comparison of generator torque in the full load region condition

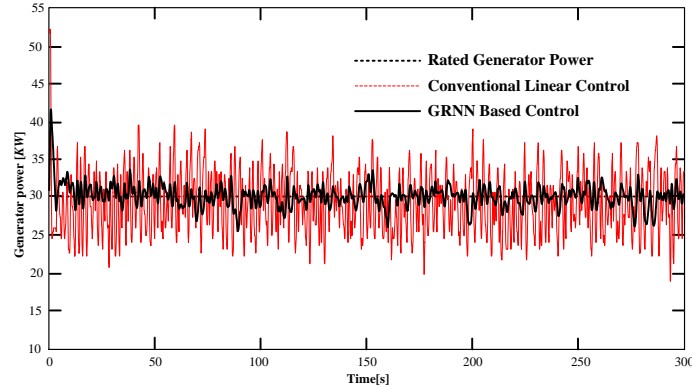

Fig. 7 (d). Comparison of generator power in the full load region condition



# 7 Conclusion

This paper presents a hybrid adaptive control approach for the VS-VP half-direct driven WECS by combining pitch angle control with variable generator torque regulation in different operating regions. Both the generator torque and pitch angle are controlled in partial or full load regions. Furthermore, an effective nonlinear PID pitch controller is proposed to track the pitch

5 angle reference command in the full load region of operation. The effectiveness of the proposed approach and the novel pitch controller are verified based on comparative simulations which validate that the proposed method is much faster, more accurate and effective than the conventional linear approach.

## Appendix.

Test parameters of the WECS

| Parameter | Description | Value |
|---|---|---|
| $\rho$ | Air density | 1.25 kg/m$^3$ |
| $J_r$ | Wind rotor inertia | 30 kgm$^2$ |
| $K_r$ | Wind rotor external damping | 2.6×10$^5$ Nm/rad |
| $K_g$ | External damping of generator | 1.6×10$^5$ Nm/rad |
| $N$ | Gearbox ratio | 43 |
| $R$ | The stator winding resistance | 1.5 Ω |
| $L_d\,L_q$ | The d, q axis stator inductance | 5.3 mH |
| $K_g$ | The generator torque constant | 0.68 Nm/A |
| $\xi$ | Learning rate of GRNN | 0.1 |
| $\alpha_0\,\alpha_1\,\alpha_2$ | Nonlinear interval indexes of $f\,(\varepsilon,\alpha,\delta)$ | 0.8, 0.6, 0.1 |

10 **Acknowledgements**

The authors would like to thank the Science Fund for Creative Research Groups of National Natural Science Foundation of China under Grant No.51221004.



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
