# Peer review of "Hybrid adaptive control for variable-speed variable-pitch wind energy systems using general regression neural network"

_Wind Energy Science, 2018_

## Referee Comment (RC1) · Anonymous Referee #1 · 27 Jun 2018

This paper presents a novel hybrid adaptive control approach for the variable speed-variable pitch (VS-VP) semidirect driven WECS by combining pitch control with variable generator torque regulation in different operating regions. This paper has some interesting aspects and merits, but needs major revisions. 1. In the introduction section, the literature review is rather old, the authors need to update this by adding new literature. 2. The WECS modelling section is so common, the authors need to eliminate some content to make this section concise. 3. The control strategies section also has the same problems, the authors need to address this problem. 4. Please outline the main new contributions of your paper. 5. The test results seem interesting, the authors may need to present more about this test.

---

## Referee Comment (RC2) · Anonymous Referee #2 · 5 Jul 2018

The concept that is presented is to let a GRNN decide a pitch angle set point which is then smoothed, compared to the current demanded pitch angle and a new set point is handed to the system.

Based on the paper I cannot judge whether this approach has any merit.

Apart from the remarks of the other reviewer, I can add the following points: 1) the literature reviewed is not only 'dated' but also very narrow in its application, not including any more general control oriented literature. 2) the introduction should focus more on the actual, common state of the art and explain why the common state of the art approach is chosen. 3) Examining the presented controller behavior in somewhat

more detail, in figures 6(a) to (d) one can see that the reference controller is behaving very oddly, especially near t=125 s, where the rotor speeds up for no apparent reason, the wind is not increasing yet and the torque is approximately flat. This leads me to conclude that the reference controller is either tuned very poorly or implemented badly. Similarly the power of the reference controller in full load seems to be very irregular. The GRNN shows mildly better behavior but would be by definition outperformed by a conventional controller using constant power control. The problem seems to be the rotorspeed, which must vary strongly (but is not presented) with a fairly high frequency, the cause of which is unclear. 4) throughout the paper there are claims that the new approach is better(e.g. in the introduction, "much faster, more accurate and effective", but I find no substantial proof that this is indeed the case.

All in all, the underlying work may have merit, but the authors do not present this in a way that this is plausible.

---

## Editor Comment (EC1) · G. J. W. van Bussel (Editor) · 10 Jul 2018

Dear Author,

I hope you have read the comments of both anonymous reviewers. Before I will take a final decision I would like to see your response on the comments of the reviewers.

Best Regards

Gerard van Bussel

---

## Referee Comment (RC3) · Anonymous Referee #3 · 23 Jul 2018

The paper could be more concise when it presents conventional modeling principles, but more in detail on the test results. It is still questionable at this point whether the new control approach is "much faster" and "more accurate", because the baseline control strategy might be badly tuned. It needs to be shown that the conventional controller is tuned as well as it can be, before it can be shown that the novel neural network approach has significant benefits.

Perhaps it is also important to give more details on the learning convergence, which may impact practical applicability.